# Single-molecule analysis reveals self assembly and nanoscale segregation of two distinct cavin subcomplexes on caveolae

Yann Gambin[1], Nicholas Ariotti[1], Kerrie-Ann McMahon[1], Michele Bastiani[1], Emma Sierecki[1], Oleksiy Kovtun[1], Mark E Polinkovsky[1], Astrid Magenau[2,3], WooRam Jung[1], Satomi Okano[1], Yong Zhou[4], Natalya Leneva[1], Sergey Mureev[1], Wayne Johnston[1], Katharina Gaus[2,3], John F Hancock[4], Brett M Collins[1], Kirill Alexandrov[1]*, Robert G Parton[1,5]*

[1]Institute for Molecular Bioscience, The University of Queensland, Brisbane, Australia; [2]Center for Vascular Research, University of New South Wales, Sydney, Australia; [3]Australian Centre for Nanomedicine, University of New South Wales, Sydney, Australia; [4]Department of Integrative Biology and Pharmacology, The University of Texas Medical School at Houston, Houston, United States; [5]Centre for Microscopy and Microanalysis, The University of Queensland, Brisbane, Australia

*For correspondence:
k.alexandrov@uq.edu.au (KA);
r.parton@imb.uq.edu.au (RGP)

**Competing interests:** The authors declare that no competing interests exist.

**Reviewing editor**: Suzanne R Pfeffer, Stanford University, United States

**Abstract** In mammalian cells three closely related cavin proteins cooperate with the scaffolding protein caveolin to form membrane invaginations known as caveolae. Here we have developed a novel single-molecule fluorescence approach to directly observe interactions and stoichiometries in protein complexes from cell extracts and from in vitro synthesized components. We show that up to 50 cavins associate on a caveola. However, rather than forming a single coat complex containing the three cavin family members, single-molecule analysis reveals an exquisite specificity of interactions between cavin1, cavin2 and cavin3. Changes in membrane tension can flatten the caveolae, causing the release of the cavin coat and its disassembly into separate cavin1-cavin2 and cavin1-cavin3 subcomplexes. Each of these subcomplexes contain 9 ± 2 cavin molecules and appear to be the building blocks of the caveolar coat. High resolution immunoelectron microscopy suggests a remarkable nanoscale organization of these separate subcomplexes, forming individual striations on the surface of caveolae.

## Introduction

Caveolae are an abundant feature of the plasma membrane of many vertebrate cells. The surface of adipocytes, endothelial cells, smooth muscle, skeletal muscle and many other cell types is characterized by a dense covering of these small invaginations with a characteristic striated coat, as viewed by electron microscopy, and by the presence of membrane proteins termed caveolins (*Peters et al., 1985*; *Kurzchalia et al., 1992*; *Rothberg et al., 1992*; *Way and Parton, 1995*; *Scherer et al., 1996*; *Parton and Del Pozo, 2013*). Three caveolins are present in mammalian cells with caveolin-1 (CAV1) and caveolin-3 (CAV3) essential for caveolar formation in nonmuscle and muscle cells respectively. Caveolins bind cholesterol and fatty acids and form homo-oligomers required for caveolar formation. Approximately 140 CAV1 molecules associate with a single caveola in mammalian cells (*Pelkmans and Zerial, 2005*) and in a model prokaryotic system upon caveolin expression (*Walser et al., 2012*). Genetic ablation of caveolins in mice has diverse cellular consequences with impact upon

**eLife digest** If you could look closely enough at the surface of some animal cells, especially fat or muscle cells, you would see that they are covered with pocket-like indents called 'caveolae'. These structures are thought to help the cells communicate with the outside world, but they can also be used by viruses to gain entry into living cells.

Examining these caveolae even closer would reveal that these pockets contain proteins called caveolins that bind to each other—and also to cholesterol and fatty acids—to form a scaffold that help to maintain the shape of the caveolae from inside the cell. Each caveolae in a mammalian cell typically contains over 100 caveolin proteins. Caveolar coat proteins, or cavins for short, are also important building blocks for caveolae: however, we know relatively little about the interactions between caveolins and cavins.

Now, Gambin et al. have used powerful new single-molecule techniques to study these interactions. These experiments looked at the three main types of cavin proteins that associate with caveolae, and by tracking individual protein molecules they showed that cavin1 can interact with either cavin2 or cavin3, but that cavin2 and cavin3 do not interact with each other. Furthermore, cavin2 and cavin3 exist in separate stripes on a caveolae. Gambin et al. also stretched the cell membrane by forcing cells to take in extra water, and showed that this caused the cavin coat to peel away from the caveolae and break down into distinct cavin1-cavin2 and cavin1-cavin3 building blocks.

Faulty versions of caveolins and cavins have both been associated with several diseases in humans, including heart disease and muscle disorders. As such, an improved understanding of the formation and break down of caveolae may prove useful for developing treatments for these conditions.

numerous signal transduction pathways and lipid dysregulation. Human patients lacking CAV1 show a severe lipodystrophy while CAV3 mutations, many of which disrupt caveola formation in muscle, are associated with a number of muscle diseases.

Recent years have seen a dramatic increase in our understanding of caveolae with the characterization of a family of caveolar coat proteins (*Hill et al., 2008*; *Bastiani et al., 2009*; *Hansen et al., 2009*; *McMahon et al., 2009*; *Bastiani and Parton, 2010*; *Hansen and Nichols, 2010*). Cavin1/polymerase transcript release factor (PTRF), cavin2/SDPR (serum deprivation response protein), cavin3/SRBC (serum deprivation response factor-related gene product that binds to C-kinase) and cavin4/MURC (muscle-restricted coiled-coil protein) are cytoplasmic proteins characterized by conserved putative N-terminal coiled coil domains (*Hansen and Nichols, 2010*). Cavin1 was originally identified as a nuclear protein that can dissociate paused ternary transcription complexes (*Jansa et al., 1998*), while cavin2 (*Gustincich and Schneider, 1993*) and cavin3 (*Izumi et al., 1997*) were identified as protein kinase C (PKC) substrates and have been suggested to function in the targeting of PKC to caveolae. Cavin4 is predominantly expressed in cardiac and skeletal muscle (*Ogata et al., 2008*; *Tagawa et al., 2008*). The cavin proteins have been shown to co-associate and form a cytosolic complex(es) in cells lacking CAV1 (or CAV3), but are recruited to the cell surface to stabilize caveolae in cells expressing CAV1. Ablation of cavin1 expression causes loss of caveolae with CAV1 being released into the bulk membrane whereas expression of cavin1 or cavin2 in cells lacking cavins but expressing endogenous CAV1 is sufficient to generate caveolae (*Hill et al., 2008*; *Hansen et al., 2009*). Like caveolins, cavins have also now been linked to many disease conditions including cardiomyopathies, lipodystrophy, and skeletal muscle disorders (*Parton and Del Pozo, 2013*).

Caveolae possess a unique cytoplasmically facing striated coat enriched in CAV1 (*Rothberg et al., 1992*). Given the recent identification of the cavin cytoplasmic coat, essential for the formation and regulation of caveolae, one could hypothesize that CAV1 would associate with the cavin complex and aid in generating the striated coat. Furthermore, this cavin coat may provide a possible mechanism for spatial and temporal regulation of caveola formation: caveolae form at the plasma membrane as caveolins and cavins associate, rather than earlier in the exocytic pathway (*Hill et al., 2008*; *Hayer et al., 2010*). In addition, the dissociation of the cavin coat complex could potentially provide a mechanism to disassemble caveolae. This may be crucial for caveolar function in setting membrane tension and in mechanosensing as increased membrane tension causes caveolar flattening and dissociation of

cavin1 (*Sinha et al., 2011*). However, the mechanisms underlying the formation of the cavin complex(es), their stoichiometry and association with caveolae are all unknown.

Here we have developed new methods that allow reconstitution of the cavin complex and performed a quantitative assessment of cavin complex formation. We use single-molecule fluorescence for its proven ability to directly observe multiple populations and quantify interactions in complex mixtures. These techniques are especially well suited for the study of coat proteins and have been used to study the mechanisms of clathrin assembly/disassembly (*Böcking et al., 2011*). Those studies required labeling of recombinantly expressed purified proteins, which in the case of the cavin complex is difficult (*Hansen et al., 2009*). We have taken an alternative approach to obtain the stoichiometry of the cavin complex and the interactions between the members of the cavin family (cavin1, cavin2, cavin3) directly from cell extracts. We could observe fascinating behaviour of the cavin members, with exquisite segregation of interactors and defined stoichiometries in mixed oligomers. By combining these experiments with novel electron microscopy techniques we show the surprising existence of two distinct cavin complexes, cavin1-2 and cavin1-3. Remarkably, the two complexes co-interact with individual caveolae but associate within distinct striated nanodomains.

## Results

### Single-molecule analysis of fluorescently tagged cavins expressed in mammalian cells

Formation of higher order structures through oligomerization is a common behaviour of proteins involved in control of membrane dynamics. Here we set out to determine what role oligomerization of cavins plays in the biogenesis of caveolae. To this end we sought a technique that would allow rapid and quantitative analysis of homo- and hetero- interactions of proteins in vivo and in vitro. We chose single-molecule fluorescence spectroscopy as a way of directly assessing interactions between cavin proteins in complex mixtures.

Fluorescence spectroscopy at the single-molecule level typically requires labeling with specific organic dyes, sufficiently bright for detection of individual molecules. The brightness of genetically encoded fluorophores such as the widely used eGFP and mCherry is considered too low to allow their individual detection, especially when proteins are freely diffusing and not immobilized on a surface. As a result, while the actual single-molecule experiment consumes typically only a few thousands of molecules, a typical sample preparation requires preparative protein expression, purification and labeling. This limits the throughput of the technology and biases its application to proteins that can be obtained in an active form through recombinant expression in *Escherichia coli*. This is typically not possible with cavin proteins, which show a tendency to degrade upon expression in a bacterial host (*Hansen et al., 2009*).

In order to resolve this bottleneck, we performed single-molecule analysis of proteins tagged with fluorescent domains such as eGFP and mCherry expressed in mammalian cells. To test this experimentally we expressed GFP- and Cherry-tagged cavin1 proteins in MCF-7 cells. As MCF-7 cells are cavin and caveola-deficient the expressed cavin1 should be mostly in the cytosol. The transfected cells were mechanically lysed, centrifuged to remove large membranous material and the supernatant was analyzed using a confocal microscope configured for single-molecule detection.

The principle of the technique and the results obtained are schematized in *Figure 1*. In brief, the brownian motion of freely diffusing proteins brings them in and out of the confocal detection volume created by two lasers that simultaneously excite the GFP and the Cherry fluorophores. When fluorescent proteins transit through the focal volume the emission of GFP and Cherry fluorophores is detected simultaneously on two separate single-photon counting detectors. The measurement generates a highly temporally resolved (100 ns) fluorescence time trace, binned in 100 ms time intervals to obtain sufficient signal/noise ratio. As shown in *Figure 1C* the signal obtained for co-expressed cavin1-GFP and cavin1-Cherry shows coincident bursts of fluorescence with large amplitudes. The presence of the two fluorophores at the same time in the focal volume demonstrates that the cavin1 proteins self-interact. However, these data alone are not sufficient to distinguish between a dimer and higher order structures. In order to extract this information, each burst of fluorescence can be analysed for intensity of GFP and Cherry signals and duration of the burst. The former analysis is based on the assumption that multiple fluorescent proteins are brighter than individual GFP/Cherry. The detailed analysis of the amplitude of each burst should allow us to calculate the number of proteins in a complex. In the burst

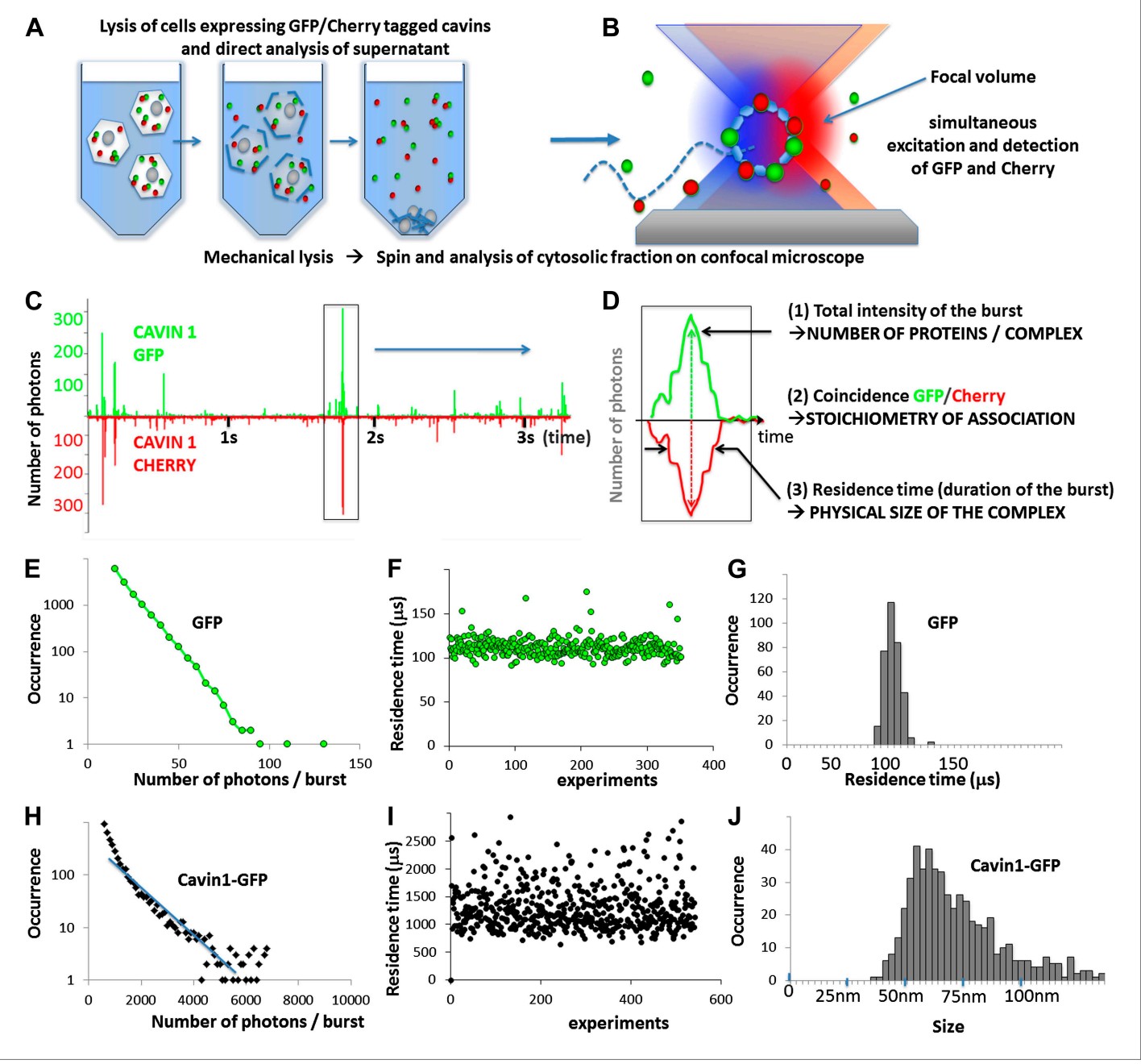

**Figure 1**. Principles of single-molecule detection and analysis of cavin1. (**A**) Sample preparation for single-molecule analysis of fluorescently tagged proteins expressed in mammalian cells. Cells transfected with GFP and Cherry fusions of the cavin1 are mechanically lysed and centrifuged to remove cellular debris and the supernatant is placed in a chamber for single-molecule detection. (**B**) Schematic representation of single-molecule fluorescence experiment in which the proteins freely diffuse in and out of the focal volume created by two lasers simultaneously exciting the GFP and Cherry fluorophores. (**C**) An example of a single-molecule trace obtained for lysates of the cells co-expressing cavin1-GFP and cavin1-Cherry. The numbers of photons detected in green and red channels are plotted as a function of time. The trace shows simultaneous bursts in both GFP and Cherry channels that reflect formation of complexes containing both fluorophores. (**D**) Detailed analysis of a fluorescence burst from (**C**). Each fluorescent burst is analysed for three parameters: (1) the coincidence between the GFP and Cherry fluorescence that reflects co-diffusion of at least two proteins with different tags; (2) the total brightness of the burst, indicating the number of proteins present in the oligomer; (3) the burst profile that is determined by the rate of diffusion and reflects the apparent size of the complex. (**E**) Histogram of the number of photons measured per burst for GFP expressed in MCF-7 cells. The distribution of bursts is consistent with the behaviour of a monomeric GFP. The data was used to calibrate the brightness profile and estimate the number of cavins-GFP molecules. (**F**) GFP residence time in the confocal volume plotted against the number of observations; (**G**) a histogram of data shown in (**F**) that reveals a very tight distribution of residence times in the focal volume (or diffusion times) around 110 µs. This value is

*Figure 1. Continued on next page*

Figure 1. Continued

used to convert the observed residence time into apparent size in the subsequent experiments. (**H**) A plot of burst size distribution of cavin1-GFP expressed in MCF-7 cells. The high quantum yield of the observed particles suggests that cavin1 forms oligomers approximately 50 times brighter than a single GFP molecule. (**I**) as in (**F**) but for cavin1-GFP. The duration of the burst for cavin1-GFP is also much higher than that of GFP indicating that the apparent size of the diffusing objects is at least 10-fold larger (**J**) as in (**G**) but X-axes represents size as calculated using residence time of GFP as a reference. The distribution of cavin1 sizes is broad and is centered on 60 nm.

duration analysis we expect time-trajectories of proteins to be different for small and larger oligomers, as large protein complexes will diffuse slower and spend more time in the focal volume. This is quantified burst-by-burst by correlating the fluorescent signal intensity as done in Fluorescent Correlation Spectroscopy (FCS) ('Materials and methods') and allows extraction of residence times in the focal volume for proteins or protein oligomers.

In order to calibrate the brightness and duration of GFP/Cherry fluorescent bursts, we analysed the time trace obtained for GFP (*Figure 1E*). Here we observed much smaller fluorescent bursts than for cavin1. The diffusion of the molecule in the focal volume is random, and the optimal trajectory maximizing the numbers of photons emitted is exponentially rare. The burst profile decreases rapidly and GFP molecules yield a maximum of 100 photons (see 'Materials and methods' for the explanation of the burst profile plots). We performed the same brightness analysis for Cherry monomers, and adjusted the excitation laser intensity to obtain the same burst profile as for GFP and a maximum of 100 photons per Cherry monomers. This ensures accurate 'counting' in both GFP and Cherry channels, correcting for the differences in extinction coefficients, quantum yields and detection efficiency of the two fluorophores. Using these brightness values as references we estimated that cavin1 forms complexes that contain 50 ± 5 proteins.

We next analyse the apparent size of the cavin1 oligomer. As in the earlier case, the measurement is calibrated by GFP, monomeric at the concentration used (100 pM), which displays a well-defined residence time in the focal volume of 100 μs. The residence times measured for cavin1 oligomers are approximately 10 times slower than GFP. The structure of GFP is well known and the near-spherical β-barrel has a typical diameter of 5 nm. If we assume that the cavin1 oligomer forms an isotropic complex and if we simplify its shape to a sphere, we can estimate the diameter of the diffusing object to approximately 60 nm (*Figure 1J*). These data demonstrate that cavin1 spontaneously assembles into a higher order structure of defined size even in the absence of the scaffolding membrane protein CAV1.

## Mapping interactions between the cavin family proteins

Although cavin1 and CAV1 appear to be sufficient to form caveolae, cells generally express 3 cavin family members that all associate with caveolae (*Bastiani et al., 2009*; *Bastiani and Parton, 2010*; *Hansen and Nichols, 2010*). After the observation that cavin1 can form a large oligomeric structure, we next analyzed the ability of the other cavins to form homo- and hetero-oligomers.

We performed dual colour co-transfection of MCF-7 cells and single-molecule coincidence analysis of all three cavins. As shown in *Figure 2A,B*, in the cytosol of MCF-7 cells cavin1 forms homo-complexes but also stable heteromeric complexes when expressed with cavin2 or cavin3. Strikingly, we could not detect any mixed oligomers when cavin2 and cavin3 were co-expressed. *Figure 2C* demonstrates that co-expressions lead to two segregated populations of cavin2-cavin2 and cavin3-cavin3 oligomers.

In order to analyse further the stoichiometries of association between cavin members, we needed to tune the co-expression levels of the different cavin proteins in a more controlled manner. We chose to produce them in vitro, using cell-free expression as it enables the precise control over the ratios of multiple co-expressed proteins via titration of DNA template concentrations. We used a recently developed *Leishmania tarentolae*–based cell-free system (LTE) as it is derived from a eukaryotic organism and is capable of producing complex eukaryotic proteins in functional form (*Mureev et al., 2009*; *Kovtun et al., 2011*).

We first compared the properties of cavin1-oligomers produced in LTE cell-free system to the ones previously observed in MCF-7 cells. The open format of cell-free protein expression enables real time observation of protein production, folding and interactions. To obtain the time resolved picture of cavin complex assembly, LTE was loaded directly into the single-molecule observation chamber mounted on the confocal microscope and primed with DNA template coding for cavin1-GFP (*Figure 2—figure supplement 1*). After 2 hr of expression, the sizes and brightness of cavin1-GFP assemblies were very

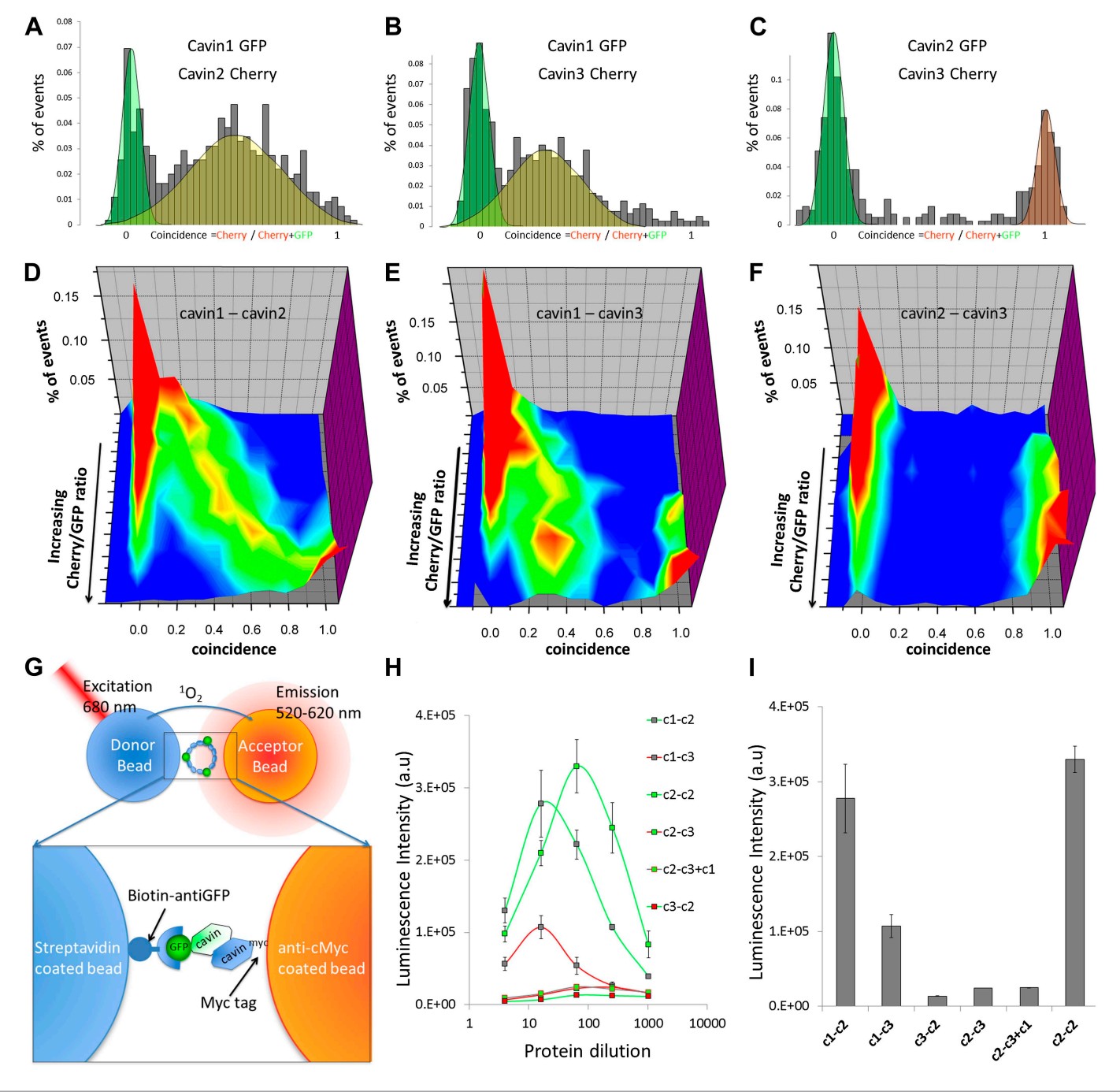

**Figure 2**. Single-molecule coincidence and ALPHAScreen mapping of interactions among cavin1, 2 and 3. (**A**) Histogram of single-molecule coincidence between cavin1-GFP and cavin2-Cherry co-transfected in MCF-7 cells. The coincidence is calculated as the ratio of intensity in the Cherry channel divided by the sum of the signals in the GFP and Cherry channels. GFP-only bursts show a coincidence at 0 and Cherry-only oligomers are located at coincidence = 1. For the oligomers containing both fluorophores, the coincidence ratio is a measure of the stoichiometry of the assembly. (**B**) Same as (**A**), coincidence between cavin1-GFP and cavin3-Cherry. (**C**) Same as (**A**), coincidence between cavin2-GFP and cavin3-Cherry. (**D**–**F**): using cell-free protein expression, three-dimensional histograms of single-molecule coincidence between GFP and Cherry cavin proteins at expression ratios spaning from 100% GFP to 100% Cherry. For each DNA ratio of cavin-GFP and cavin-Cherry, we collected >1000 bursts and plotted the corresponding histograms of coincidence. We varied the ratio of cavins and created a stack of 10 histograms representing the various stoichiometries. The histograms for individual pairs were then aggregated into 3D plots. (**D**) Evolution of mixed oligomers of cavin1-GFP and cavin2-Cherry revealing formation of oligomers with a full range of stoichiometries. (**E**) The cavin1 and cavin3 plot shows formation of cavin1-cavin3 oligomers with predominantly 3/1 composition.
*Figure 2. Continued on next page*

*Figure 2. Continued*

(**F**) The cavin2 and cavin3 plot shows that these proteins do not form mixed oligomers. (**G**) Schematic representation of ALPHAScreen principle. This bead–bead assay relies on transfer of singlet oxygen from a donor bead to a luminescent acceptor bead when protein–protein interactions bring the beads within 200 nm (see 'Materials and methods' and SI for details). (**H**) A plot of ALPHAScreen signal across the concentrations of cavin proteins attached to donor and acceptor beads. The interactions for cavin2 and cavin3 in the presence or absence of cavin1 display amplitudes close to the background signal. (**I**) Values obtained for cavin1-GFP and cavin2-myc, cavin1-GFP and cavin3-myc, cavin2-GFP and cavin2-myc reveal robust interactions. However the curves obtained for cavin3-GFP and cavin2-myc, cavin2-GFP and cavin3-myc demonstrate that cavin2 and cavin3 cannot bind to each other. The triple co-expression of cavin2-GFP, cavin3-myc and untagged cavin1 results in no change in binding, suggesting that cavin1 cannot act as a bridge between cavin2 and cavin3.

The following figure supplements are available for figure 2:

**Figure supplement 1**. Single-molecule fluorescence trace of cavin1-GFP during expression in the cell-free system.

**Figure supplement 2**. Comparison of cavin1 oligomers observed in MCF-7 cells and expressed in the cell-free system.

**Figure supplement 3**. Principle of the ALPHA screen.

**Figure supplement 4**. Pull-down analysis of the cavin complex formation in MCF-7 cells.

---

similar to the ones obtained for MCF-7 cells (*Figure 2—figure supplement 2*), indicating that we could use the cell-free system to reproduce the behaviour of cavin oligomerization. The time-trace of cavin production contains additional information. Surprisingly, we found that most of the cavin1 protein assembled into oligomers within the first 15 min while the concentration was still below 10 nM (*Figure 2—figure supplement 1*). This indicates that cavin oligomer assembly is a very rapid process that operates spontaneously at nanomolar concentrations.

In the next step we used the in vitro expression system to analyse the interactions of cavin1 with cavin2 and cavin3. We observed that the composition of oligomers formed by co-expression of cavin1 and 2 varied with levels of expression resulting in continuous transition from cavin-1 to cavin-2 homo-oligomer (*Figure 2D*). On the contrary, we found that only a maximum of ca. 30% of cavin3 could be incorporated into cavin1 oligomers (*Figure 2E*). Similarly to MCF-7 cells, we found complete segregation between cavin2 and cavin3 oligomers regardless of co-expression ratio (*Figure 2F*).

Available biological data strongly indicate that in vivo all three cavin members are present on a single caveola (*Bastiani and Parton, 2010*). The observed segregation between cavin2 and cavin3 was so unexpected that we decided to use an alternative biophysical method to analyse interactions between cavin2 and 3. In order to detect even weak but potentially relevant biological interactions we employed ALPHAScreen (Amplified Luminescent Proximity Homogeneous Assay Screen). This sensitive bead-based proximity assay (*Figure 2G*) is able to detect interactions in a wide range of affinities (from pM to mM) (*Eglen et al., 2008*). When used for protein–protein interaction analysis the approach typically involves purified proteins (*Waller et al., 2010*; *Mackie and Roman, 2011*; *Demeulemeester et al., 2012*). In our case, we analysed interactions of protein pairs directly after their co-expression in the cell-free system and utilized GFP and myc tags for capture of the proteins on the reporter beads (see 'Materials and methods', *Figure 2—figure supplement 3*; *Sierecki et al., 2013*). As shown in *Figure 2H*, co-expression of cavin1 and 2 or cavin1 and 3 resulted in a very strong positive signal. However, no interaction between cavin2 and 3 was detected using this approach. This was confirmed by classical biochemistry and pull-downs from MCF-7 cells as shown in *Figure 2—figure supplement 4*. Taken together these results indicate that cavin2 and 3 do not interact with each other, either in vitro or in vivo. This is a surprising finding that could potentially indicate that cavin1 operates as a bridging factor between cavin2 and 3. To test this idea we repeated the ALPHAscreen assay where we measured the binding of cavin2 and 3 under conditions of cavin1 co-expression. *Figure 2I* shows that the presence of cavin1 did not enhance the interaction of cavin2 and 3. To further corroborate this finding we performed single-molecule coincidence analysis where co-expression titration of unlabeled cavin1 to a fixed ratio of cavin2-GFP/cavin3-Cherry was monitored. The total fluorescence of GFP and Cherry cavin2/3 decreased as more cavin1 was co-expressed reflecting competition of the templates for the translational machinery. However no mixed oligomers containing both cavin2 and cavin3 were produced (data not shown). To verify this result in mammalian MCF-7 cells, we next expressed cavin2-Cherry

together with cavin3-GFP in the presence of non-fluorescent cavin1-FLAG. Supporting our in vitro findings we observed that the vast majority of the cavin2 and cavin3 proteins do not interact even when cavin1 is co-expressed (*Figure 3A*). These findings suggest the existence of previously undetected segregation in cavin oligomers.

## Assembly of the cavin coat on caveolae

The experiments described above revealed the intrinsic propensity to assemble into defined structures even in the absence of the scaffolding membrane. We next set out to characterize a more complete cellular system that included CAV1 protein to target cavins to the membrane. Hence, we expressed Cherry-tagged cavin1 in MCF-7 cells expressing CAV1-GFP. Microscopic analysis revealed that cavin1 was now located at the plasma membrane but a fraction of CAV1-labeled structures decorated with cavin1 proteins were found in the cytoplasm, presumably reflecting the dynamic nature of caveolae in cultured cells (*Pelkmans and Zerial, 2005*; *Boucrot et al., 2011*). This allowed us to extend our single-molecule analysis of fluorescent caveolar proteins to analysis of caveolae in the post nuclear supernatant of transfected cells.

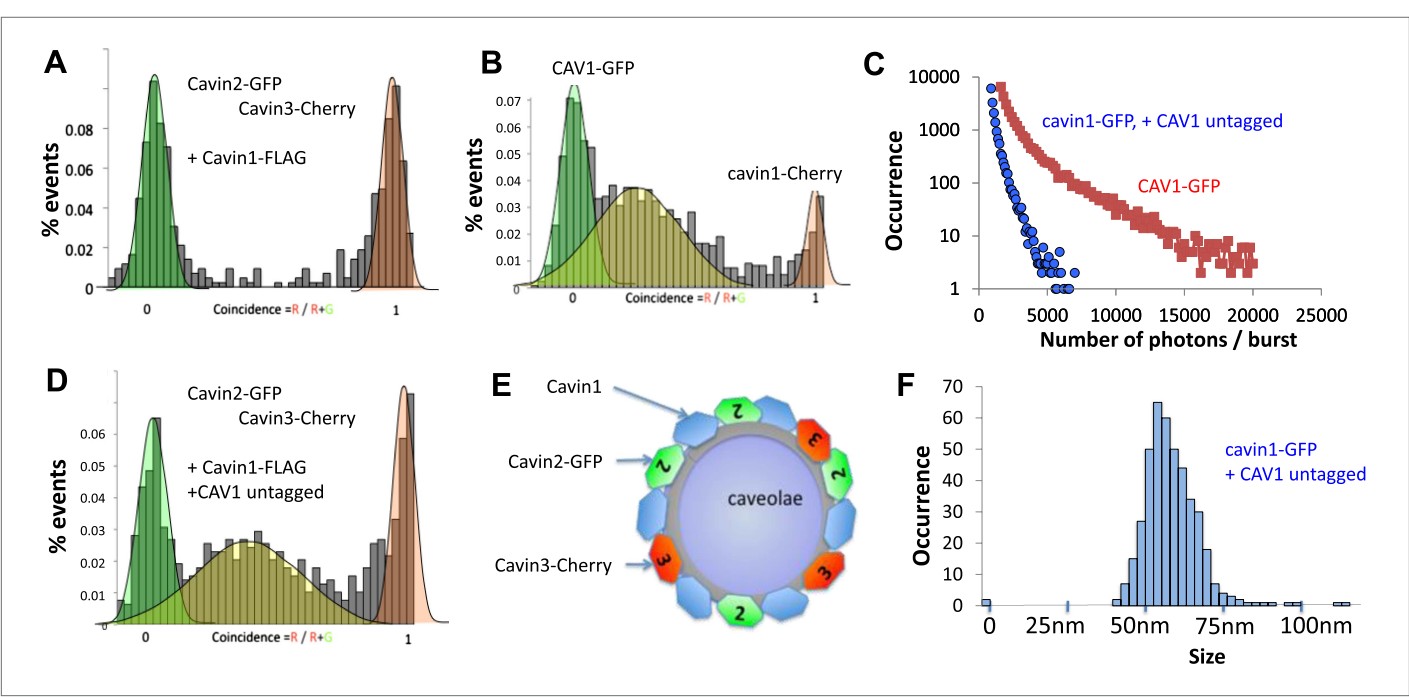

**Figure 3**. Single-molecule counting and coincidence analysis of cavin oligomerization in MCF-7 cells in the presence of CAV1. (**A**) Histogram of single-molecule coincidence between cavin2-GFP and cavin3-Cherry in MCF-7 cells co-expressing cavin1. The lack of coincidence (0.25<coincidence<0.75) demonstrates that cavin2 and cavin3 do not interact even in the presence of cavin1. (**B**) Histogram of single-molecule coincidence between CAV1-GFP and Cavin1-Cherry. The distribution shows a large peak of coincidence centered around C = 0.25, indicating that cavin1 localizes on caveolae with approx. three CAV1 molecules for one molecule of cavin1. (**C**) Distribution of burst brightness from MCF-7 cells transfected with CAV1-GFP (shown in red) or cavin1-GFP and CAV1 untagged (in blue). The brightness of CAV1-GFP is typically 3.5 higher than the brightness of cavin1-GFP. (**D**) Histogram of single-molecule coincidence between cavin2-GFP and cavin3-Cherry in MCF-7 cells when both cavin1 and CAV1 are co-expressed. (**E**) Schematic representation of the co-localization of cavin2 and cavin3 when cavin1 and CAV1 are co-expressed, indicating that the two subcomplexes are assembled on the caveolae. (**F**) Histogram of apparent single-molecule sizes of cavin1-GFP co-expressed with CAV1 (untagged). The size distribution is centered around the size of caveolae, suggesting that the cavin1 complex wraps around caveolae without significantly increasing its apparent size.

The following figure supplements are available for figure 3:

**Figure supplement 1**. Effect of co-transfection of CAV1 in MCF-7 cells on the size and brightness of cavin1-GFP oligomers.

**Figure supplement 2**. Single-molecule coincidence analysis of cavin interactions in HeLa cells.

**Figure supplement 3**. Localization of Cavin 2-Cherry and Cavin 3-GFP in MDCK cells.

We first characterized the behaviour of CAV1-GFP present in the cytoplasm. We observed intensely fluorescent puncta in the supernatant of CAV1-GFP transfected cells. As shown in *Figure 3C*, the intensity of these puncta reaches values that are 150 to 200-fold higher than the brightness of a single GFP. As caveolae contain up to 180 CAV1 proteins, this corresponds well to the expected brightness of fully-assembled caveola (*Pelkmans and Zerial, 2005*). As shown in *Figure 3B*, we performed dual-colour coincidence experiment between cavin1-Cherry and CAV1-GFP. The presence of a large coincident peak shows that most of the cavin1-Cherry co-diffuses with CAV1-GFP. As cavin1 does not associate with CAV1 in non-caveolar cellular compartments, such as when CAV1 is moving along the secretory pathway or if caveolae are disassembled (*Hill et al., 2008*; *Hayer et al., 2010*) we are confident that these represent bona fide caveolae. The coincidence peak is centred between C = 0.25 and C = 0.3, indicating that each caveola contains three to four times more CAV1 than cavin1. This observation corroborates the brightness analysis that demonstrates that the CAV1 signal is 3.5 times brighter than that of cavin1 (*Figure 3B*). Both methods suggest that the number of cavin1 per caveolae is estimated, again, between 40 and 50.

The number of cavin1 subunits per oligomer did not differ significantly between CAV1 positive and negative cells. However, while the apparent number of subunits per cavin1 assembly has not significantly changed, the presence of CAV1 had a pronounced effect on the distribution of particle sizes (*Figure 1J*; *Figure 3F*, *Figure 3—figure supplement 1*), narrowing the distribution to around 60 nm, which corresponds to the typical size of a caveola.

To understand how the observed segregation of cavin subcomplexes translates into formation of native caveolae we repeated our cellular experiments with the full complement of caveolae-forming molecules. As in the experiments described above, we co-expressed cavin2-Cherry and cavin3-GFP together with CAV1 and cavin1-FLAG in MCF-7 cells. We observed coincidental diffusion of cavin2 and cavin3 indicating that they were part of the same assembly (*Figure 3D*). The brightness and physical size of the cavin2-cavin3 complex correspond to a mixed coat of cavins formed on a single caveola. These data suggest that cavin1, cavin2 and cavin3 are found together at the surface of the same caveolae, as depicted in *Figure 3E*. Identical results were obtained when cavin2-Cherry and cavin3-GFP were transfected into HeLa cells that express endogenous CAV1 and cavin1 (*Figure 3—figure supplement 2*). The co-binding of cavin2 and cavin3 on the same caveolae fits well with their colocalisation observed in MDCK cells, which also express endogenous CAV1 and cavin1 (*Figure 3—figure supplement 3*).

## Hypo-osmotic treatment induces dissociation of the cavin coat into two subcomplexes

The results described above suggest that the cavin1 proteins have an intrinsic propensity to assemble into a large complex in the absence of the CAV1/caveolae scaffold. A single complex made of only 50 cavin1 proteins would be sufficient to wrap around a caveola, possibly as a loose mesh. However it is possible that the cavin1 complex produced in the absence of their CAV1 membrane target represents an artificial dead-end product. We therefore took advantage of hypo-osmotic treatment to dissociate the caveolae-associated cavin1 complex from the plasma membrane and to study the properties of the released native cavin complex.

We co-transfected MDCK cells with cavin1-GFP and cavin1-Cherry and performed cell swelling experiments by subjecting cells to hypo-osmotic treatment for 20 min. Cytoplasmic fractions were prepared from cells before and after the treatment and were subjected to single-molecule analysis. The dual-colour experiment showed that the cavin1 oligomer recovered in the cytosolic fraction diffuses much faster than the caveolae bound protein: the apparent size of the cavin1 complex is reduced to ‹ 30 nm (calculated as diameter of a diffusing spherical object) (*Figure 4A*). Yet the protein remained oligomeric, as all fluorescent events register both fluorophores (*Figure 4B*). The distribution of burst brightness showed that the observed oligomers are composed of 9 ± 2 cavin1 proteins (*Figure 4C*). This suggests that the cavin coat is actually made of multiple stable subcomplexes that can be released from the caveolae without being disassembled into monomers. Because the cavins have such a high propensity to oligomerise, it is possible that the subcomplexes would form in the cytosol before reaching the CAV1-rich membrane, and that the subcomplexes can reversibly assemble on a caveola.

The hypotonic treatment mediated release of cavin protein from the membrane provides an opportunity to test the cohesion of the observed cavin1, 2 and 3 caveolar coat. Examination of the cavins released in the cytosol of hypo-osmotic cells showed that all cavins now existed as small oligomers.

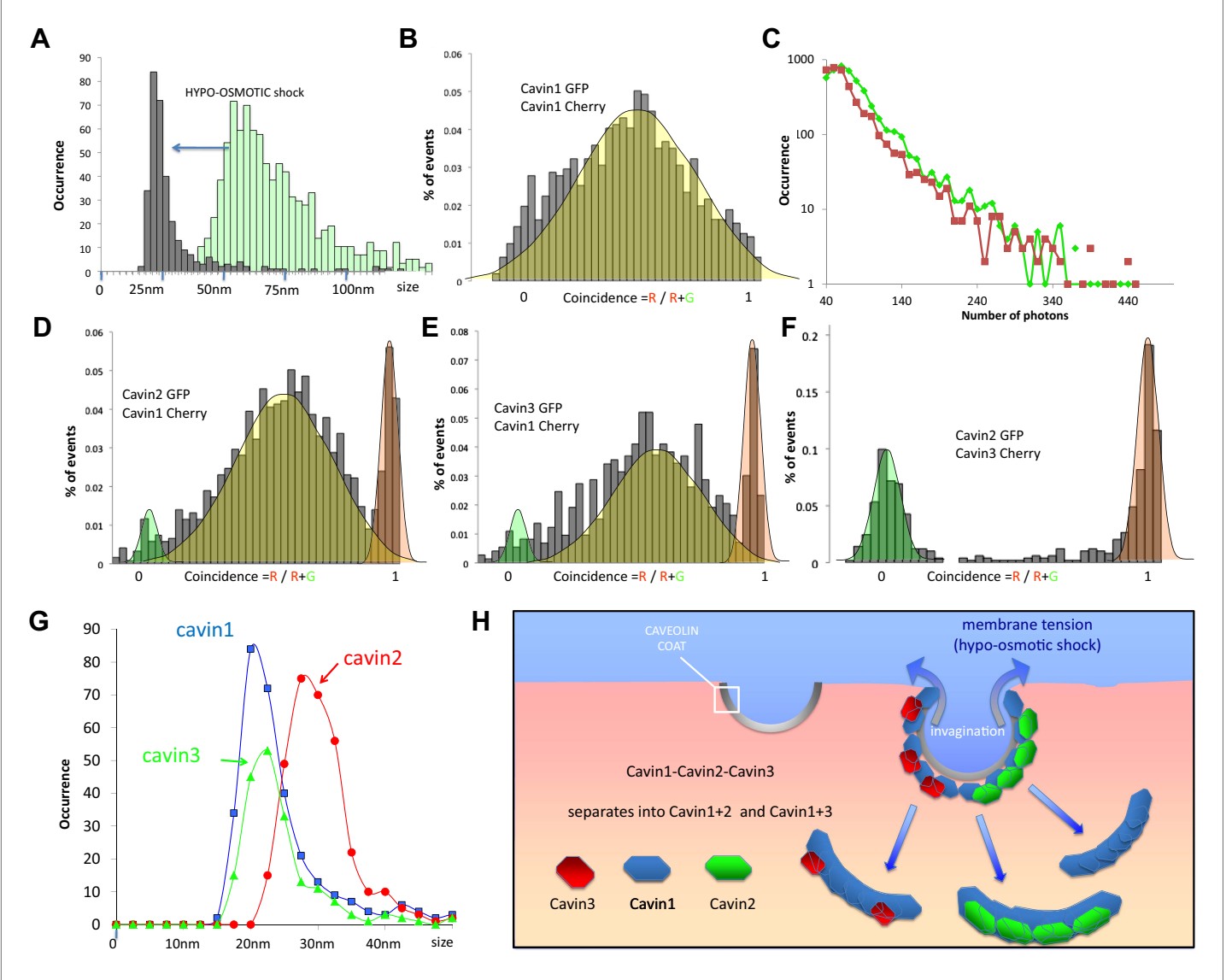

**Figure 4**. Dissociation of the cavin coat upon cell swelling induced by hypo-osmotic treatment. (**A**) Histogram of apparent single-molecule sizes for cavin1-GFP before (green) and after (grey) membrane stretching induced by hypo-osmotic treatment. The size of observed cavin1 oligomers decreases from 60 nm to 20 nm. (**B**) Histogram of single-molecule coincidence for cavin1-GFP and cavin1-Cherry after hypo-osmotic treatment. We observe that solubilized cavin1 remains oligomeric as all bursts contain both GFP and Cherry fluorophores. (**C**) Burst brightness distribution for cavin1-GFP (green) and cavin1-Cherry (red) in oligomers released from the membrane upon hypo-osmotic treatment. Oligomers contain GFP and Cherry in equal amounts; the total fluorescence indicates that the sub-oligomers are typically made of 8–10 cavin1 proteins and hence significantly reduced in size compared to the isotonic conditions (*Figure 1*). (**D**) Histogram of single-molecule coincidence between cavin2 and cavin1 after hypotonic treatment. Data show that all cavin2 are bound to cavin1 but approximately 15% of the oligomers contain cavin1 only. (**E**) Histogram of single-molecule coincidence between cavin3 and cavin1 after membrane stretching. Data show that all cavin3 are bound to cavin1 and approx. 20% of cavin1-only oligomers are observed. (**F**) Histogram of single-molecule coincidence between cavin2 and cavin3 after membrane stretching. The absence of coincidence suggests that while cavin2 and cavin3 can localize to the same caveolae, their release from the membrane during stretch causes them to separate into two different subcomplexes. (**G**) Histogram of apparent single-molecule sizes obtained after hypo-osmotic treatment for cavin1-GFP, cavin2-GFP and cavin3-GFP, in the presence of co-expressed cavin1-Cherry. The measurements indicate that the cavin1-cavin1 and cavin1-cavin3 subcomplexes are very similar in size (average of 20 nm), but the apparent size of the cavin1-cavin2 subcomplex is higher with an average of 30 nm. (**H**) Model of cavin1-cavin1, cavin1-cavin2 and cavin1-cavin3 subcomplexes dissociation from CAV1 domains upon membrane stretching mediated by cell swelling.

The following figure supplements are available for figure 4:

**Figure supplement 1**. Pull-down analysis of the effect of membrane stretching on the association between cavin members.

Remarkably we observed a complete segregation of cavin2 and cavin3. We observed rare oligomers (>15%) made of cavin1-only (red peak in histograms), but not oligomers made of cavin2-only or cavin3-only. The single-molecule coincidence histogram shows clearly that cavin3 remains bound to cavin1, and cavin2 bound to cavin1, and that these two subcomplexes detach separately from the membrane. This was confirmed by classical biochemistry and pull-downs as shown in *Figure 4—figure supplement 1*. Interestingly, even the size of the two subcomplexes differs, with cavin1-cavin1 and cavin1-cavin3 forming oligomers of similar size (20 nm) that are significantly smaller than the cavin1-cavin2 sub-complex (30 nm). The complete lack of co-diffusion indicates that subcomplexes do not form direct interactions in solution; it also suggests that on the membrane, they do not bind strongly to form stable mixed structures.

### Electron microscopy demonstrates cavin2 and cavin3 co-localize to the same caveola but form segregated striated nanodomains

The identification of two distinct cavin complexes raised the question of their assembly and spatial organization on the cytoplasmic face of caveolae. An immuno-EM method was developed using an 'unroofing' protocol modified from *Heuser (2000)*. 3T3-L1 fibroblasts transiently expressing cavin1-Cherry, cavin2-Cherry (co-transfected with cavin1-FLAG) or cavin3-mCherry (co-transfected with cavin1-FLAG) were unroofed and pre-embedding labeling was performed using a directly conjugated 3 nm-α-RFP gold (α-RFP antibody is cross-reactive with the Cherry-tag) as described in 'Materials and methods'. Cavin1 (*Figure 5A*–top two rows), cavin2 (*Figure 5A*–middle two rows) and cavin3 (*Figure 5A*–bottom two rows) respectively demonstrated immunolabeling consistent with a striated pattern. Predicted three-dimensional orientations of the immunolabeling were generated from thin sections and are depicted in *Figure 5* right hand columns.

To determine if subcomplexes of cavin2 and cavin3 could be resolved on intact caveolae, immuno-EM was performed on the basolateral surface of the plasma membrane in cells expressing cavin2 and cavin3. 3T3-L1 fibroblasts were transfected with either GFP- or Cherry-tagged cavin constructs and were 'unroofed'. Pre-embedding labeling was performed using with two different directly conjugated antibodies: 3 nm-α-RFP and 7 nm-α-GFP. In accordance with the single-molecule fluorescence data, cavin1-Cherry (3 nm gold–coloured red) and cavin3-GFP (7 nm gold–coloured green) demonstrated a close spatial association and co-labeled the same caveolae with no perceivable separation (*Figure 5B*). Dual-labeling of cavin2-GFP (7 nm–coloured green) and cavin3-Cherry (3 nm–coloured red), in cells expressing roughly equivalent amounts (with co-expression of cavin1-FLAG), demonstrated that cavin2 and cavin3 were localized to the same caveola (*Figure 5C*). However, within an individual caveola significant spatial separation between these proteins was observed, consistent with separate striations. In order to verify and quantify these observations, Ripley's K function bivariate analysis of the co-clustering of cavin2-GFP and cavin3-Cherry to the co-clustering of cavin1-Cherry and cavin3-GFP was performed. There was a significantly lower level of co-clustering of cavin2-cavin3 as compared to cavin1-cavin3 (*Figure 5D*).

These results, together with single-molecule fluorescence studies, demonstrate that cavin1-cavin2 and cavin1-cavin3 complexes form sub-caveolar complexes on the cytoplasmic face of caveolae. These subcomplexes form distinct striations that we propose can shape caveolae and peel away as caveolae flatten (see model *Figure 5E*).

## Discussion

Here we report the development of a novel single-molecule fluorescence method for analysis of self-assembly and caveolar association of cavin proteins in vivo and in vitro. We quantitatively describe the homo- and hetero-oligomerization properties of the three cavin family members and we demonstrate a specificity of interactions that generate two discrete subcomplexes. All three cavin members can associate with the same caveolae, but in two subcomplexes that can dissociate separately in response to membrane stretch.

For each of the cavins the resulting assembly is of a defined size, estimated to be approximately 50 monomers. The consistent size of the cavin oligomers shows that the process is not random and that the cavins have an intrinsic property to limit their growth. One possible mechanism for this behaviour would be a structure regulated by curvature, driven by higher order interactions between the cavin subunits.

The methodological toolbox developed in this study allowed us to quantitatively examine the cavin complexes in three different states: (1) in association with caveolae in cell extracts and on intact caveolae at the plasma membrane; (2) in the cytoplasm, in the absence of caveolin and caveolae; and (3) upon dissociation from membrane-bound caveolae in response to hypotonic medium.

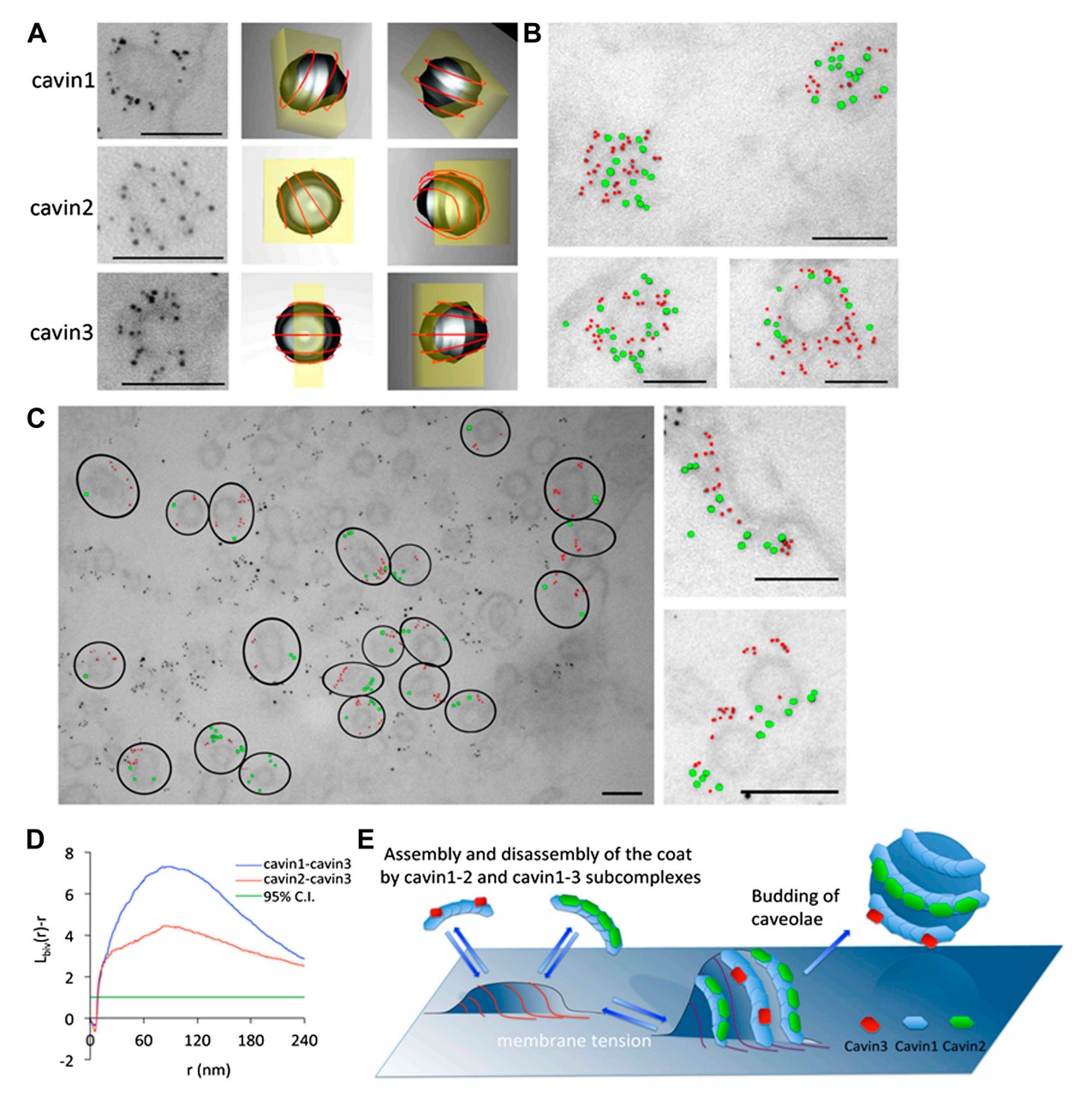

**Figure 5**. The cavins form part of the striated caveola coat and cavin2 and cavin3 localize to the same caveola while remaining spatially distinct. (**A**) High magnification images of immuno-EM labeled 3T3-L1 fibroblasts expressing cavin1 (top row), cavin2 (middle row) and cavin3 (lower row) and their predicted three-dimensional orientation (right hand columns, red line = predicted orientation based on a model of cytoplasmic caveolar striations, yellow box = field of view from electron micrograph of 60 nm section). Scale bars, 100 nm. (**B**) High magnification images of cavin1-Cherry (3 nm gold–colored red) and cavin3-GFP (7 nm gold–colored green) label the same caveolae and have significant overlap. Scale bars, 100 nm. (**C**) Cavin-2-GFP (7 nm gold) and cavin-3-Cherry (3 nm gold) demonstrates the spatial separation between both proteins within individual caveolae, consistent with the formation of separate striations. Black circles in the low magnification image denote the spatial limit of the individual caveola and its associated coat of caveolae that are positive for both cavin2 and cavin3. Scale bars, 100 nm. (**D**) Bivariate clustering analysis of different plasma membrane lawns expressing various combinations of cavin proteins. Significantly higher co-clustering of cavin1-cavin3 was observed when compared to cavin2-cavin3 (CI = confidence interval). Scale bars, 100 nm. (**E**) Model of association and dissociation of the cavin1-cavin2 and cavin1-cavin3 subcomplexes.

1) Our data suggest that the cavin complex associated with caveolae in the cell extract is made of typically 50 cavin proteins. We believe that the contribution of endogenous proteins must be minor because of the striking consistency in the size of the complexes observed both for cavins and caveolins

in our different systems (*Leishmania* cell-free system, MCF-7 cells that lack endogenous cavin proteins, and MDCK cells that natively express cavins). The comparison of results obtained in *in vitro* and in vivo systems shows clearly that each of the cavin proteins has an intrinsic property for self-association to form stable oligomers. After synthesis, in vitro large oligomers are formed at very low concentration (<10 nM, see *Figure 2—figure supplement 1*), suggesting that cavins would form the same oligomers at endogenous levels. Most importantly, recent studies using biochemical approaches (*Ludwig et al., 2013*) show striking agreement with our observations. A ratio of CAV1 to cavin of 4:1 determined biochemically (*Ludwig et al., 2013*) is in good agreement with the brightness analysis and CAV1-cavin1 coincidence data presented here. Similarly, the ratio of cavin1 to cavin3 of ~2:1 is in excellent agreement with the result obtained from quantitative pull-downs (*Ludwig et al., 2013*). The same ratio of 2:1 of cavin1 to cavin2 observed in subcomplexes released from caveolae (*Figure 4D*) is similar to that reported in *Ludwig et al. (2013)* but notably in the absence of caveolae more variable ratios are observed (*Figure 2* in MCF-7 and cell-free systems). The studies indicate that the number of cavin proteins in the caveolar coat is relatively low, considering the number of CAV1 proteins within a caveola, and compared to the other protein coats such as clathrin (made of 30 + triskelias, approximately 200 subunits).

We find that when caveolae are disassembled by membrane stretch, cavins are released as subcomplexes of ~9 cavin molecules, suggesting a typical caveola contains typically 5 cavin subcomplexes. Cavin1 is believed to form trimers, possibly through a coiled-coil domain (*Ludwig et al., 2013*), suggesting that each sub-complex may be composed of three trimers. Our data clearly shows however, that cavin2 and cavin3 are not present in the same assemblies with cavin1: the sub-complexes are mutually exclusive for cavin2 or cavin3. Furthermore, cavin2 and cavin3 also segregate spatially on the surface of caveolae as revealed by high resolution immunoelectron microscopy. This segregation is consistent with the formation of striations as shown in *Figure 5*. Thus stable units of cavin1-cavin2 and cavin1-cavin3 can come together on the membrane to regulate the formation of the curved caveolar structure.

2) Even in the absence of the scaffolding element CAV1, cavins can assemble in the cytoplasm forming a complex that has a similar size to a caveola assembly. But this does not mean that the cavin complex will be fully formed before interacting with caveolae. Expression in cells lacking caveolins may reflect an artificial situation as cavins and caveolins are generally expressed together in vivo. We believe the large complexes formed under these conditions may be rare in vivo, possibly explaining the slow association of expressed cavin with newly-arrived caveolin at the plasma membrane (25 min) when fluorescently-tagged cavin is expressed in mammalian cells (*Hayer et al., 2010*). The size of the cavin1 complex in the absence of CAV1 shows a broader size distribution and can be even larger than the size of a cavin1-CAV1 containing particle, where the presence of CAV1 appears to condense the complex (*Figure 3F* vs *Figure 1J*, raw data presented in *Figure 3—figure supplement 1*). This observation suggests that the 50-mer cavin1 complexes form a more heterogeneous population in terms of their size and shape, which become more tightly organized when bound to CAV1. The fact that oligomers of cavin1 form so readily in the cell-free expression system (*Figure 2—figure supplement 1*) suggests that the subcomplexes could be pre-assembled rapidly in the cytosol, before reaching the CAV1-rich domains at the membrane. We hypothesize that a combined interaction with membranes (*Hansen et al., 2009*) and caveolin itself will result in recruitment of the subcomplexes to the budding caveolae, where they can be assembled into the striated structures observed on intact caveolae (this study and *Ludwig et al., 2013*).

3) The use of hypotonic treatment to swell cells allowed us to gain insights into the complexes that dissociate from membrane-bound caveolae in a reversible fashion (*Sinha et al., 2011*). We show that the cavin coat dissociates into stable sub-elements, cavin1/cavin2 and cavin1/cavin3 subcomplexes with a well-defined size of 9 ± 2 cavin proteins. Interestingly, based on the observed residence times in the focal volume the complexes have slightly different sizes with cavin1-cavin1 20 nm, cavin1-cavin2 30 nm and cavin1-cavin3 20 nm. These sizes are calculated based on the assumption that the diffusing objects are spherical, which remains an oversimplification until further structural information is available. The cavin sub-oligomers are very stable and the proteins remain associated even after caveola disassembly, with few monomers detected in solution, as shown by the distribution of sizes; all cavin proteins are associated with objects of apparent size greater than 15 nm (*Figure 4G*). By high resolution immuno-EM and quantitative spatial analysis we show for the first time that cavin2 and cavin3 label distinct subdomains of single caveolae. The preferential labeling of spatially-distinct linear elements raises the possibility that the cavin1/cavin2 and cavin1/cavin3 complexes actually make up the caveolar striations seen by high resolution SEM and deep etch electron microscopy (*Peters et al., 1985*;

*Rothberg et al., 1992*) consistent with EM observations with miniSOG-cavin fusion proteins (*Ludwig et al., 2013*). These striated 'nanodomains' could be released from ('peel off') the caveolae/membrane in response to membrane stretch. We speculate that this could potentially release two distinct subcomplexes, each able to reach distinct cellular destinations or interact with different partners. These interactions could be further regulated by additional post-translational modifications to individual cavins. An additional possibility is that the properties of individual caveolae can be modulated by their ratio of cavin1, 2, and 3 subcomplexes to vary their individual properties, including stability or morphology.

The mode of assembly of cytoplasmic caveolar coat proteins shown here is distinct from that of any other known coat complex. Cavins form characteristic homo- and hetero-oligomeric complexes, which associate to form striated nanodomains that can wrap around caveolae. In contrast to other coat proteins, such as clathrin, which form a very dense basket-like network in a cooperative and stepwise process through protein–protein interactions and binding to membrane components and then is disassembled into individual triskelia in an ATP-dependent process, we hypothesize that caveolar assembly relies on association of pre-oligomerized cavin subcomplexes that give rise to the unique caveolar striations. Further functional characterization of the cavin subcomplexes should prove illuminating in understanding the diverse roles of caveolae.

## Materials and methods

### Cell culture and constructs
MCF-7 and MDCK cells were maintained as previously described (*Kirkham et al., 2008*). MDCK cells were grown in DMEM/F-12 (Life Technologies, Carlsbad, CA, USA) supplemented with 10% FBS and 2 mM L-glutamine. MCF-7 cells were grown in DMEM supplemented with 10% FBS and 2 mM L-glutamine. Caveolin-1, cavin1, cavin2 and cavin3 constructs were cloned as described in *Hill et al. (2008)*. Tagged constructs were transfected using Lipofectamine 2000 reagent (Life Technologies, CA, USA) following the manufacturer's instructions using a 1:3 ratio of DNA:Lipofectamine. 3T3-L1 fibroblasts were grown in DMEM with 10% fetal bovine serum (FBS) and 1 mM L-glutamine (Life Technologies, CA, USA). All transient transfections were performed with Lipofectamine 2000 (Life Technologies) as per the manufacturer's instruction.

### Cell-free protein expression
Cell-free expression of proteins for interaction mapping used the eukaryotic Leishmania cell-free system. Manufacture and supplementation of lysate from *Leishmania tarentolae* is as described in *Kovtun et al. (2011)*. DNA templates for the various ORFs used the Gateway cloning system, with ribosome engagement with T7 transcribed mRNA mediated by the species independent translation initiation site (*Mureev et al., 2009*). Coupled transcription/translation occurred for 3 hr at 27°C unless described as otherwise.

### Single-molecule spectroscopy
Single-molecule spectroscopy was performed based on (*Gambin et al., 2009*, *2011*). 20 µl of samples are used for each experiment, placed into a custom-made silicone 192-well plate equipped with a 70 × 80 mm glass coverslip (ProSciTech Australia). Plates were analysed on a Zeiss LSM710 microscope with a Confocor3 module, at room temperature. Two lasers (488 nm and 561 nm) are co-focussed in solution using a 40 × 1.2 NA water immersion objective (Zeiss, Germany); fluorescence was collected and split into GFP- and Cherry-channel by a 560 nm dichroic mirror. The GFP emission was further filtered by a 505–540 nm band pass filter and the Cherry emission was filtered by a 580 nm long-pass filter.

The single-molecule multicolour detection method is based on a simple principle: the two excitation lasers are focussed within the same point, creating a very small observation volume (~1 fl). The proteins are tagged with genetically-encoded GFP and Cherry fluorophores. Proteins diffuse freely by Brownian motion, and they enter and exit the confocal volume constantly. To reach single-molecule detection, the samples were diluted to approximately 100 pM concentration, so that only single proteins or protein complexes are present in the confocal volume.

For single-molecule coincidence, the time-trace of GFP and Cherry intensity was binned in 100 microseconds time 'bins'. That is, we calculate the number of photons collected in 100 ms. When the fluorescence is greater than the defined threshold (20 photons), we consider that fluorophores are present in the focal volume. When a burst lasts for more than one time bin, we 'stitch' the bursts together. That is, we calculate the total fluorescence emitted by the diffusing objects as long as

they are in the focal volume. For single-molecule coincidence, the coincidence ratio is calculated as *Mukhopadhyay et al. (2007)* [Intensity(Cherry)−leakage from GFP channel]/total intensity (Intensity (Cherry)+Intensity(GFP)); the leakage from GFP to Cherry channel was experimentally determined at 10%. An average of 1000 events was accumulated for each single-molecule coincidence histogram.

The brightness analysis is calibrated using GFP expressed in the same MCF-7 cells or cell-free extracts. Due to random diffusion, most of the events are the result of incomplete transfer through the focal volume, and few bursts represent the maximal number of photons that the GFP-tagged proteins can emit.

The perfect trajectory, maximizing the time spent in the middle of the focal volume, is extremely rare, and the protein complex is more likely to exit the detection volume quickly. The escape from the detection volume can be modeled as a first order decay: the probability of a long residence in the focal volume is the combined effect of a succession of random events, each with a successful outcome. In this simple model, following the laws of probability, the longer bursts will be exponentially less frequent.

The probability of detecting a burst of a given brightness (plotted in *Figures 1 and 3*) seems to be described well by this model, as shown by the linear decrease in the log-scaled graphs. The behaviour of GFP is extremely well defined, as we obtained a profile of burst sizes that rapidly decreases, and no bursts were observed above 100 photons. In case of larger oligomerization of the GFP- labeled cavins, multiple proteins and their fluorescent labels pass the confocal volume at the same time, resulting in an even higher photon count.

For single-molecule measurement of residence times, the time trace was acquired with a much higher frequency of 100 ns per time bin. We define a 1 s window before and after each burst and the intensity I(t) at short timescales is analysed for correlation as $<G(T)>=<I(t).I(t + T)>/<I(t)>.<I(t + T)>$ using the FCS (Fluorescence Correlation Spectroscopy) software (Zeiss710 Confocor) (*Ferreon et al., 2009*). This single-molecule FCS gives direct information on the size of protein complexes. FCS typically leads to an average correlation curve and an average diffusion value, measured over thousands of proteins. On the contrary, the analysis burst-by-burst gives access to a distribution of sizes, where large aggregates can be separated from smaller oligomers.

This dwell time measured at the single-molecule level for GFP monomers corresponds perfectly to the one measured at a 10 nM concentration using FCS. We did not attempt single-molecule FCS using Cherry fluorophores, as the FCS curves typically display an important contribution from triplet states with a characteristic time of 30 µs. This phenomenon interferes with the accurate determination of residence times.

## ALPHAScreen assay

ALPHAScreen cMyc detection and Proxiplate-384 Plus 384-wells plates were purchased from Perkin Elmer (MA, USA). Proteins (one bearing a N-terminal eGFP tag, the other labeled with C-terminal mCherry-Myc) were co-expressed in the cell-free system by adding mixed DNA vectors in 10 µl of the *Leishmania tarentolae*-based cell-free system (to a final DNA concentration of 30 nM for the GFP-vector and 60 nM for the Cherry-vector) The mixture was incubated for 3.5 hr at 27°C. Four serial dilutions of the proteins of ¼ were made in buffer A (25 mM HEPES, 50 mM NaCl). The AlphaScreen Assay was performed in 384-well plates. Per well, 0.4 µg of the Anti-Myc coated Acceptor Beads (PerkinElmer, MA, USA) was added in 12.5 µl reaction buffer B (25 mM HEPES, 50 mM NaCl, 0.001% NP40, 0.001% casein). 2 µl of the diluted proteins and 2 µl of biotin labeled GFP-nanotrap (diluted in reaction buffer A to a concentration of 45 nM) were added to the acceptor beads, followed by incubation for 45 min at room temperature. Then 0.4 µg of Streptavidin coated donor beads diluted in 2 µl buffer A were added, followed by an incubation in the dark for 45 min at room temperature. ALPHAScreen signal was recorded on a PE Envision Multilabel Platereader, using the manufacturer's recommended settings (excitation: 680/30 nm for 0.18 s, emission: 570/100 nm after 37 ms). The signals were then averaged and normalized to background.

## Hypo-osmotic treatment

For hypo-osmotic treatments, cells were treated with 100% DMEM (iso-osmotic) or with a dilution of DMEM (1:10) in sterile water. The duration of the hypo-osmotic treatment was 20 min at 37°C.

## Sample preparation, GFP trap and western blotting

MCF-7 or MDCK cells were transfected with equal amounts of cavin constructs as specified in the pertinent figure legends. The soluble cytoplasmic fractions were prepared by extensive washing of cells in PBS. Cells were then scraped into PBS containing protease (Merck Pty Ltd, Kilsyth, Australia)

and phosphatase (Roche Diagnostics Australia, Castle Hill, Australia) inhibitors. Cells were mechanically disrupted by syringe lysis using a 1 mL syringe and a 25-gauge needle (Terumo, Tokyo Japan). Cells were then pelleted at 2000 rpm, 10 min at 4°C. 1/120th of the supernatant was retained as the starting material. The remaining supernatant was mixed with 20 µl of prewashed sepharose conjugated antiGFP beads (*Kovtun et al., 2011*) for 30 min at 4°C on a rotating wheel. The beads were pelleted and washed three times in PBS supplemented with protease and phosphatase inhibitors and were boiled in 4 X SDS containing sample buffer at 75°C for 2.5 min to preserve the fluorescence of GFP. Sample were pelleted at maximal speed for 10 min at room temperature and the supernatant was separated by SDS-PAGE and the fluorescence was detected and quantified in the gel using the BioRad ChemiDoc MP Imaging System (BioRad, Hercules, California, USA).

## Electron microscopy

Cells were grown as described above, plated onto 6 cm tissue culture dishes (TRP), transfected overnight with the plasmid of interest, trypsinized (TryplExpress–Gibco) and replated onto 3 cm tissue culture dishes (TRP) that had been pre-coated with Poly-L-Lysine (Sigma–Aldrich, St Louis, MO, USA). Cells were incubated for 4 hr to ensure attachment and flattening of cells to the coated dish, washed with PHEM buffer (80 mM PIPES, 25 mM HEPES, 10 mM EGTA and 2 mM $MgCl_2$ at pH 6.9) and sonicated with a probe tip sonicator (Vir Sonic Ultrasonic Cell Disruptor 100–setting 3) at an angle of 45°. Baso-lateral membranes were selected using fluorescence microscopy (EVOS$_{fl}$ AMG-Life Technologies, CA, USA fitted with green, red and blue filters). Areas with fluorescent membranes but lacking nuclei were selected and washed repeatedly with PHEM buffer, blocked in 0.2% bovine serum albumin and 0.2% fish skin gelatin then incubated with α-GFP and α-RFP antibodies. Dual labeling was performed as follows: unroofed cells were labeled first with the large 7 nm α-GFP antibody for 30 min, washed with block and then re-labeled with the 3 nm α-RFP antibody for 30 min. Basal membranes were then fixed in 2.5% glutaraldehyde (Electron Microscopy Sciences, PA, USA) for 1 hr at room temperature and then post-fixed in 1% $OsO_4$ for 1 hr, also at room temperature. Dishes were serially dehydrated in increasing percentages of ethanol in a BioWave microwave and LX-112 resin was infiltrated at ratios of 1:2, 1:1, 2:1 (resin to ethanol) and 100% resin twice in a BioWave microwave. Resin was polymerized at 60°C overnight. Ultrathin sections (55–65 nm) were then cut, such that only the very first sections were collected (i.e., the most basal region of the cell) and imaged in an JEOL1011 electron microscope at 80 kV equipped with a Morada Soft Imaging System 4K × 4K camera at twofold binning under the control of iTEM (Olympus, Japan).

## Ripley's K-function analysis

Ripley's bivariate K-function analysis was performed as described (*Prior et al., 2003*).

## Additional information

### Funding

| Funder | Grant reference number | Author |
|---|---|---|
| Australian Research Council Future Fellowships | FT0991611 | Brett M Collins |
| Australian Research Council Future Fellowships | FT110100478 | Yann Gambin |
| Australian Research Council Future Fellowships | FT100100027 | Kirill Alexandrov |
| National Health and Medical Research Council (NHMRC) Project Grant | APP1025082 | Yann Gambin, Kirill Alexandrov |
| NHMRC Fellowships | 569542, 1058565 | Robert G Parton |
| NHMRC Program Grants | 511005, 1037320 | Katharina Gaus, Kirill Alexandrov, Robert G Parton |
| ARC Discovery Project Grants | DP120101298 | Robert G Parton, Brett M Collins |

| Funder | Grant reference number | Author |
|---|---|---|
| ARC Discovery Project Grants | DP130102396 | Katharina Gaus |
| ARC Discovery Project Grants | DP120101423 | Yann Gambin, Kirill Alexandrov |

The funders had no role in study design, data collection and interpretation, or the decision to submit the work for publication.

### Author contributions

YG, Designed and performed single-molecule analysis, Analysis and interpretation of data, Drafting or revising the article; NA, Designed and performed EM experiments, Analysis and interpretation of data, Drafting or revising the article; K-AM, Performed cell transfections, cell swelling experiments, Prepared cell lysates and performed pull-downs, Analysis and interpretation of data, Drafting or revising the article; MB, Performed cell swelling experiments and prepared lysate, Analysis and interpretation of data, Drafting or revising the article; ES, Designed and performed AlphaScreen assay, Analysis and interpretation of data, Drafting or revising the article; OK, WJ, Contributed materials and performed experiments, Drafting or revising the article; MEP, KG, JFH, Acquisition of data, Analysis and interpretation of data, Drafting or revising the article; AM, Performed SIM imaging, Analysis and interpretation of data, Drafting or revising the article; SO, Performed cell swelling experiments and prepared lysate, Drafting or revising the article; YZ, Performed Bivariate clustering analysis, Analysis and interpretation of data, Drafting or revising the article; NL, Drafting or revising the article, Contributed unpublished essential data or reagents; SM, WJ, Contributed to production and characterization of the cell-free expression system, Drafting or revising the article; BMC, Analysis and interpretation of data, Drafting or revising the article; KA, RGP, Conception and design, Analysis and interpretation of data, Drafting or revising the article

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
