## [Decision Letter]

Thank you for sending your work entitled “Single molecule analysis reveals self assembly and nanoscale segregation of two distinct cavin subcomplexes on caveolae” for consideration at *eLife*. Your article has been favorably evaluated by a Senior editor and 3 reviewers, one of whom is a member of our Board of Reviewing Editors.

The Reviewing editor and the other reviewers discussed their comments before we reached this decision, and the Reviewing editor has assembled the following comments to help you prepare a revised submission.

This manuscript reports an elegant approach using single molecule protein fluorescence to examine assemblies of proteins that comprise caveolae. The authors identify distinct subassemblies comprised of cavin 1 and 2 or cavin 1 and 3 when the proteins are analyzed in cell extracts. There is broad interest in the molecular composition of caveolae.

The major concern that arose during review is the following. One reviewer wrote:

"While much of the qualitative data was informative and well explained, the quantitative analysis was poorly explained and may have been incorrectly interpreted. (Given the explanation of the Methods, it is impossible to tell.) In general there was very little explanation about how the data presented related to the conclusions that were drawn.

1) The data in Figures 1 and 2 clearly show that all three cavins can self-associate and that cavin1 can associate with cavins 2 and 3, but cavins 2 and 3 do not co-polymerize. The data in Figure 1 also show qualitatively that polymerized cavins diffuse more slowly than monomeric GFP.

2) Problems arise with the quantitative analysis. (Minor point, but symptomatic: the time scales are missing.)

a) Figure 1 shows an exponential fall-off in the fraction of occurrences of different numbers of photons per burst. It is asserted that this is consistent with GFP monomer diffusion. The basis for this assertion is not explained.

b) The residence times shown in Figure 1 show a sharp boundary with a minimum residence time. This seems unlikely since by Brownian motion some particles should barely enter the measurement volume and then exit rapidly. Thus, it seems likely that this lower limit for residence time is due to some filtering of the data that is not explained well. The impact of this filtering on the calculated diffusion coefficients is not known.

c) These diffusion times are then converted to a size of a particle by a method that is not explained. Is the particle modeled as a linear polymer, as a sphere, or as some type of ellipsoid? What does the single size mean? I would guess that an effective radius of gyration is meant, but it would be good to know.

d) The analogy with true FCS measurements does not seem correct – again at least as explained. Usually, FCS measures the fluctuations in the number of particles in a measuring volume as a function of time. This study seems designed to measure intensity correlations as a single particle diffuses into and out of the measuring volume. There is a rigorous theory for interpreting FCS data in terms of diffusion coefficients. I do not know what is the basis for interpretation of the data in this study.

3) In the Results section, assertions are made about the number of CAV1-GFP molecules in a fluorescent particle and the association with cavin1-cherry with CAV1-GFP. What is the evidence supporting these statements, and how were the data interpreted?

4) The size for a CAV1 particle (caveola) is the same as for the cavin1 complexes, which seems surprising. This needs to be explained since the caveola also includes the cavin1.

5) Figure 3 is a cartoon, but it seems to be cited in the text as providing information about a mixed coat of cavins on a caveola.

6) Figure 5 shows an EM of caveolae stained for various cavins. It is not clear to me how these were interpreted to produce the predicted orientations on the right. “

Other comments that also need to be addressed:

7) The authors conclude they have determined the composition of caveolae yet their analyses may be entirely focused on cytosolic subassemblies rather than membrane associated caveolae. The authors are encouraged to revise the text very carefully to ensure that they are accurately describing what they are monitoring.

8) In August a paper appeared in PLOS Biology from Ben Nichols using crosslinking followed by detergent extraction to characterize the same complexes. The authors should include a discussion of how their findings corroborate or differ from those reported in that study.

9) Please clarify the basis for the statement that most cavin1 assembled into oligomers within 15 minutes.

10) The manuscript slightly hypes the novelty of the methods used (but is less novel when compared to all the variations that currently exist for FCS, FCCS, etc). In this light, the last paragraph of the Discussion should be modified. Many methods already achieve what the authors write there.

11) Please include discussion of the consequences of the presence of endogenous untagged versions of the studied proteins in cells.

---

## [Author Response]

*1) The data in*
Figures 1 and 2
*clearly show that all three cavins can self-associate and that cavin1 can associate with cavins 2 and 3, but cavins 2 and 3 do not co-polymerize. The data in*
Figure 1
*also show qualitatively that polymerized cavins diffuse more slowly than monomeric GFP*.

*2) Problems arise with the quantitative analysis. (Minor point, but symptomatic: the time scales are missing.*)

*a)*
Figure 1
*shows an exponential fall-off in the fraction of occurrences of different numbers of photons per burst. It is asserted that this is consistent with GFP monomer diffusion. The basis for this assertion is not explained*.

We now have a better description of the burst profile in the main text and in the Methods section.

Note that we did not attribute a specific number of oligomers to a given brightness for each burst, as was done in “Direct Observation of the Interconversion of Normal and Toxic Forms of a-Synuclein”, by N. Cremades, *Cell* (25 May 2012). Instead, our analysis conservatively states that the highest brightness reached by the oligomers gives an indication of the maximal number of proteins included in the oligomers. Because the profile of bursts remains exponential (linear decrease in the log-scaled graphs of Figures 1, 3 and 4), we conclude that the cavin1 oligomers are relatively homogeneous in brightness. Figure 3 showing the CAV1 data confirms this observation, as CAV1 proteins assemble into well-defined objects, caveolae. The fact that we observe brighter bursts with the CAV1, corresponding to their expected oligomerisation in caveolae, strengthen our conclusion that the cavin can form typically a 50-mer.

*b) The residence times shown in*
Figure 1
*show a sharp boundary with a minimum residence time. This seems unlikely since by Brownian motion some particles should barely enter the measurement volume and then exit rapidly. Thus, it seems likely that this lower limit for residence time is due to some filtering of the data that is not explained well. The impact of this filtering on the calculated diffusion coefficients is not known*.

As described in the Methods section, we indeed have to use a threshold to detect a burst, and the events considered correspond to trajectories passing through the middle of the focal volume. This approach proved to be valid for the GFP-only construct, as the residence times measured at single-molecule resolution matches perfectly the average GFP diffusion times measured by FCS at higher concentrations. When performed on A488-only molecules, the diffusion times were about 50 microseconds, also matching the ones observed at higher concentration using regular FCS. In the case of the cavin oligomer, the threshold does not create the lower limit for the residence times, as shown by the dissociation experiments in Figure 4. Indeed, we obtain completely different profiles for the cavin1-cavin2 and cavin1-cavin3 subcomplexes, and the sizes deduced are all smaller than the ones measured for the cavin1 assembly before dissociation.

This remark is actually extremely relevant and we now included it in the manuscript.

*c) These diffusion times are then converted to a size of a particle by a method that is not explained. Is the particle modeled as a linear polymer, as a sphere, or as some type of ellipsoid? What does the single size mean? I would guess that an effective radius of gyration is meant, but it would be good to know*.

As understood by the reviewer, the size described in the paper referred to the radius of gyration of a sphere-like object. This seems to make sense for the caveola-sized oligomer formed by cavin1; it is probably more approximate in the case of the sub-complexes. As we do not know at this stage a possible diameter of rod-like sub-complexes, we cannot determine the length of these possible oligomers. The average size indicated in the manuscript correspond to the size of a sphere, and we clarified this point in the text.

*d) The analogy with true FCS measurements does not seem correct* – *again at least as explained. Usually, FCS measures the fluctuations in the number of particles in a measuring volume as a function of time. This study seems designed to measure intensity correlations as a single particle diffuses into and out of the measuring volume. There is a rigorous theory for interpreting FCS data in terms of diffusion coefficients. I do not know what is the basis for interpretation of the data in this study*.

In fact, there is no requirement for a high number of proteins to be present in the focal volume in the mathematical theory of FCS; it is only performed as such due to signal-to-noise issues. A single protein will diffuse in the same manner, whether proteins are also present in the focal volume, and the fluctuations in its burst profile are still due to the local excitation of the laser (profile of the confocal volume). The analysis of the stacks of burst profiles (events before and after detected burst) was also performed using the Zeiss710 confocal FCS software. We found a perfect match between FCS numbers and curves obtained at various GFP dilutions.

*3) In the Results section, assertions are made about the number of CAV1-GFP molecules in a fluorescent particle and the association with cavin1-cherry with CAV1-GFP. What is the evidence supporting these statements, and how were the data interpreted*?

The number of CAV1-GFP molecules on caveolae was determined using the single-molecule brightness analysis, as shown in Figure 3. The manuscript insufficiently referred to Figure 3 and we have modified the text to clarify our statements and to describe the results leading to our conclusion.

*4) The size for a CAV1 particle (caveola) is the same as for the cavin1 complexes, which seems surprising. This needs to be explained since the caveola also includes the cavin1*.

The text contains a clarified statement in the Results. We also discuss this observation in more detail in the manuscript.

*5)*
Figure 3
*is a cartoon, but it seems to be cited in the text as providing information about a mixed coat of cavins on a caveola*.

The panel 3E is only a schematic explaining the data observed in panel 3D: the coincidence between cavin2 and cavin3 when both cavin1 and CAV1 are co-expressed suggest that all proteins are now gathered on the same caveolae. We clarified this point in the Results.

*6)*
Figure 5
*shows an EM of caveolae stained for various cavins. It is not clear to me how these were interpreted to produce the predicted orientations on the right. *“

These are simply predicted orientations based on a model of caveolar striations. We have now clarified this in the text and figure legend.

*Other comments that also need to be addressed*:

*7) The authors conclude they have determined the composition of caveolae yet their analyses may be entirely focused on cytosolic subassemblies rather than membrane associated caveolae. The authors are encouraged to revise the text very carefully to ensure that they are accurately describing what they are monitoring*.

We revised the text to specify where our measurements describe cytoplasmic components and where the results apply to membrane-bound proteins.

*8) In August a paper appeared in PLOS Biology from Ben Nichols using crosslinking followed by detergent extraction to characterize the same complexes. The authors should include a discussion of how their findings corroborate or differ from those reported in that study*.

We now discuss our results in the light of this new study; see also comment #11 below.

*9) Please clarify the basis for the statement that most cavin1 assembled into oligomers within 15 minutes*.

We now refer to Figure 2—figure supplement 1 and added an extensive description.

*10) The manuscript slightly hypes the novelty of the methods used (but is less novel when compared to all the variations that currently exist for FCS, FCCS, etc). In this light, the last paragraph of the Discussion should be modified. Many methods already achieve what the authors write there*.

We deleted the whole statement and concluded the study instead with discussion of the specificity of the cavin coat compared to clathrin and other known protein assemblies.

*11) Please include discussion of the consequences of the presence of endogenous untagged versions of the studied proteins in cells*.

We now discuss the possible impact of endogenous proteins on our results.